# Microneedles in Drug Delivery: Progress and Challenges

**DOI:** 10.3390/mi12111321

**Published:** 2021-10-28

**Authors:** Muhammet Avcil, Ayhan Çelik

**Affiliations:** Imperial Bioscience Ltd., Mocatta House, Trafalgar Place, Brighton, East Sussex BN1 4DU, UK; mavcil@imperialbioscience.co.uk

**Keywords:** microneedles, transdermal drug delivery, vaccine delivery, transdermal patch technology

## Abstract

In recent years, an innovative transdermal delivery technology has attracted great interest for its ability to distribute therapeutics and cosmeceuticals for several applications, including vaccines, drugs, and biomolecules for skin-related problems. The advantages of microneedle patch technology have been extensively evaluated in the latest literature; hence, the academic publications in this area are rising exponentially. Like all new technologies, the microneedle patch application has great potential but is not without limitations. In this review, we will discuss the possible limitations by highlighting the areas where a great deal of improvements are required. Emphasising these concerns early on should help scientists and technologists to address the matters in a timely fashion and to use their resources wisely.

## 1. Introduction

The skin is designed to perform an extensive range of jobs, and its barrier properties keep the underlying organs safeguarded from external difficulties, including physical, chemical, and microbial stresses. Using the skin as the drug administration site is an attractive option for distributing therapeutics such as vaccines, drugs, biomolecules, and difficult-to-deliver small molecules. However, the hydrophobic and lipid-rich surface layer of the skin limits the bioavailability of therapeutics. Among the available transdermal drug delivery (TDD) methods, the microneedle-mediated delivery system, which is defined as the non-invasive delivery of medications through the skin surface, has attracted interest from many research institutes and companies. The defensive, inflammatory and immunological properties of the skin make the microneedle (MN) delivery system an attractive alternative drug delivery system to address the limitations associated with conventional methods [1]. The MN delivery system, which consists of an array of submillimetre-sized needles (up to 1500 μm in length) attached to a base support, has been shown to be able to penetrate into the viable epidermis of the skin, bypassing the stratum corneum (SC), the outermost layer of the skin. In this way, the delivery of pharmaceutical ingredients becomes possible in a pain-free manner, as the MN delivery system avoids interfering with the dermal layer, which is where all nerve fibres and blood vessels are mainly located. The system has been proven as a valuable technique in delivering drug molecules with higher masses (over 500 Da) and various polarities. The therapeutic ingredients include small molecules; biomacromolecules (proteins, hormones, peptides); vaccines for SARS, MERS, and COVID-19; and genes [2]. In fact, an example of an MN-based system has progressed into phase III clinical trials (www.clinicaltrials.gov, accessed on 1 August 2021).

Although microneedle technology was originally conceptualised and patented in the 1950s [3], it took some time for the benefits of microneedles to be widely recognised. It was not until 1998 that a report was released that looked at the potential use of microneedles for vaccines [4]. Since then, the number of investigational studies on MNs has grown considerably; over 4000 patents and research articles have been presented, with the number of these still rising exponentially. In particular, there has been considerable progress in recent decades, including advances in strategies of microneedle fabrication and the assessment of MNs in clinical applications to satisfy the complex requirements in actual use. Some of the pioneering and key developments in MN research have been summarised in Figure 1. Recently, MN patches have gained rapid momentum in the cosmetic field for skin moisturizing or anti-ageing applications. Most commercialised MN patches are composed of hyaluronic acid (HA), which dissolves into the skin after administration. MNs made of HA can moisturise skin tissue and deliver actives for skin improvement via their dissolution [5,6].

The shortcomings associated with the MN system, however, should be addressed in the early stages of product development [7]. To evaluate the future direction of the field, significant developments in microneedle-based research have been highlighted, accompanied by constraints that could potentially hamper the full exploitation of the system. In this short review, special emphasis is laid on these limitations, which require a great deal of attention. Stressing these concerns early on could help scientists and technologists to address these issues in a timely manner and to use their resources wisely.

**Figure 1 micromachines-12-01321-f001:**
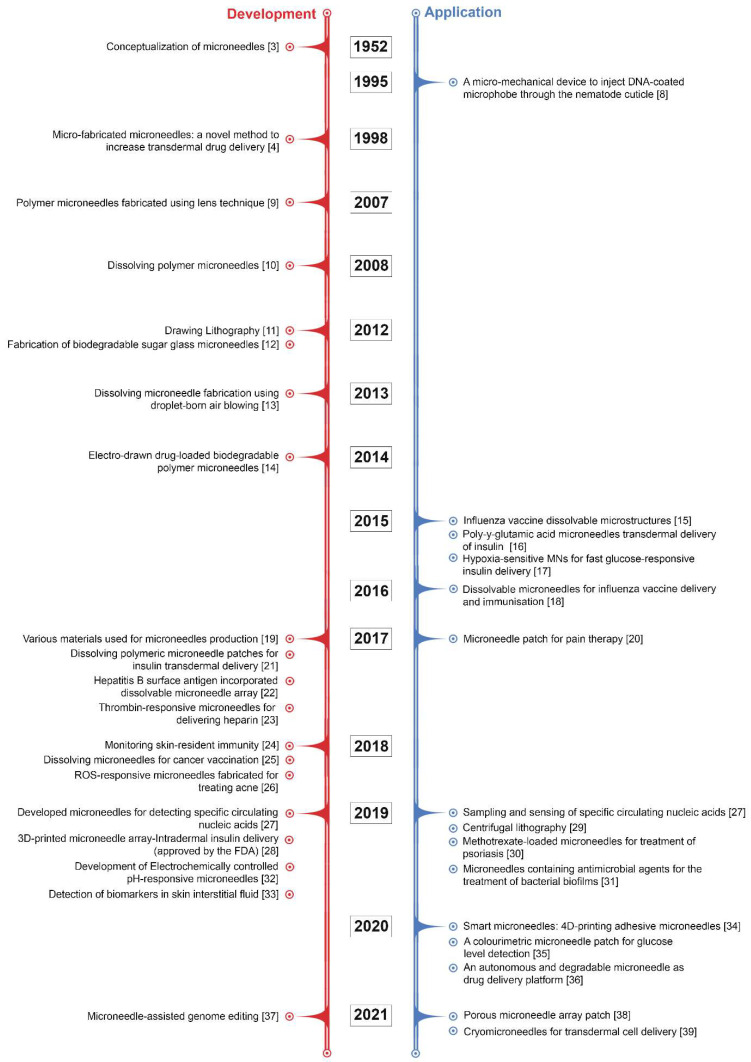
Some of the pioneering and key developments in MN research [3,4,8,9,10,11,12,13,14,15,16,17,18,19,20,21,22,23,24,25,26,27,28,29,30,31,32,33,34,35,36,37,38,39].

## 2. Microneedle-Based Delivery Approaches

The skin’s position and large surface make it a suitable and non-invasive location not only for supplying therapeutic agents but also for sampling interstitial fluid for biomarker detection. Essentially, MN-based delivery and sampling are pain-free, non-invasive, and self-administered techniques that serve as an alternative to hypodermic needles, providing enhanced patient compliance. As research activities in the field intensified in the last few decades, microneedles (MNs) are produced using various constituent materials with several designs and shapes, from metals and glass to polymers and hydrogels, in conjunction with several delivery approaches. Among the approaches, four were initially proposed [40], with another developed later [41]: poke and patch (solid MNs), coat and poke (coated MNs), poke and flow (hollow MNs), poke and dissolve (dissolving MNs), and poke and release (hydrogel-forming MNs) (Figure 2).

The pore-performing pre-treatment of the “poke and patch” approach involves the application of a solid MN patch to create small holes in the skin, followed by a conventional drug application on the surface of the skin. The first reported fabrication of solid MNs was based on silicon to deliver calcein through excised human skin in vitro [4,8]. Cost, fragility, biocompatibility, and the complex manufacturing process have steered researchers to other materials, including metals, ceramics, and polymers, in order to achieve better outcomes. Although the production of solid MNs is technically simple—no loading or coating is required—the two-step administration procedures and the no exact dosing with drug reformulations requirement are the main limitations of solid MNs, along with safety matters. Using solid MNs for the delivery of proteins, hormones, and vaccines have been reviewed in detail elsewhere [1,44].

Coating therapeutic agents on the surface of microneedles (e.g., solid MNs—metallic, silica, or polymeric) is possible to create coated MNs. This “coat and poke” approach allows for effective drug delivery provided that the formulations are stable and uniformly layered on the surface of the MNs. The drug formulation should also be water-soluble and allow layer-by-layer coating procedures. Choosing an appropriate coating technique is key for the successful generation of coated MNs. The delivery of vaccines [45], insulin [46], and hormones [47], along with other macromolecules, has been reported for the “coat and poke” approach. A further extension in applications of coated MNs has been demonstrated recently for the ultra-sensitive detection of protein biomarkers in an immunised mouse model [48]. Polystyrene microneedles coated with a primary antibody were developed to capture inflammatory biomarkers in interstitial fluid with an improved limit of detection. The main distinguishing feature of coated microneedles is their ability to avoid the degradation of bioactive molecules throughout the microneedle production process, thereby ensuring bioactivity. Furthermore, coating is one of the easiest and most controlled methods of making microneedles functional. It enables sampling and isolation, especially for microneedles with detecting capabilities. Common limitations, however, are that the small doses and loaded cargo may lessen the strength of the MNs, resulting in low strength and penetration ability.

Relatively large quantities of therapeutic ingredients may be supplied into the skin with the “poke and flow” approach, which, by using hollow MNs, could potentially overcome the dose limitation associated with solid MNs [49]. With hollow MNs, it is technically possible to control the flow and dosing by diffusion or pressure or electronically (e.g., using a pump), and to integrate them into lab-on-chip devices. Similarly, bio-macromolecules, including proteins, vaccines, mRNA, and diagnostic agents, can be delivered via hollow MNs [50,51]. These MNs can also be used for the isolation and identification of biomarkers including glucose [52], and ECG measurements [53]. Nonetheless, the construction of hollow MNs is relatively complicated and suffers from clogging, drug leakage, structural fragility, and the requirement of a larger tip diameter, which leads to poor insertion.

The “poke and dissolve” approach, in which water-soluble therapeutic agents are carried into the skin, uses mostly biocompatible/biodegradable and low-cost polymers. Hyaluronic acid, sucrose, polylactic/glycolic acid (PLA/PGA), and chitosan are among the polymers often used for the construction of dissolvable MNs (dMNs). Because of their physicochemical characteristics, which allow designing and engineering with tuneable properties and functions, biomaterials like polysaccharides have been frequently utilised to create dissolvable MNs. This has resulted in carbohydrate-based microarrays with tremendous potential for serving as an innovative step in medication administration, detection, and biological retorting [50]. A large number of articles on the production of polymeric MNs have been seen in a very short period. Unlike silicon or metal, this sort of delivery means is based on the breakdown of MNs upon exposure to the skin’s interstitial fluid. Conditional on the nature of the MN material, the dissolving process discharges the cargo from the matrix for local or systemic administration. To date, the majority of soluble MNs have been produced by utilising polymers and simple sugars and by employing casting or micromoulding methods. The therapeutic loads are encapsulated, stored, and protected in the scaffold and delivered into the targeted area after the skin insertion via a polymer erosion mechanism, without leaving any biohazardous waste. A successful application of dMNs in vaccine delivery, for instance, has been demonstrated [54]. Sucrose and fish gelatin-based MNs have been used for vaccine delivery. During the phase I trial, the influenza virus vaccines supplied by the microneedle patches (MNPs) were found to be immunogenic and safe. The drawbacks associated with the sharp waste of solid MNs, the requirement of a pump, the high cost of hollow MNs, and the sophisticated layering procedure with coated MNs are eliminated here. However, dMNs have their own drawbacks, including low mechanical strength, low doses, and doubtful penetration abilities [55].

Soft materials such as swellable polymers, including poloxamer [56], PEG-crosslinked poly(methyl vinil ether-co-maleic acid), and silk fibroin with phenylboronic acid/acrylamide [57], have recently been used for hydrogel-forming MNs. The polymers absorb the interstitial fluid into their 3D matrix upon insertion into the skin, resulting in the delivery of therapeutic agents through created micro-conduits. The response-related delivery of therapeutic applications, such as glucose-responsive insulin delivery, is particularly noteworthy, as it eliminates the need for constant glucose monitoring and relies on the responses of physiological signals. Diagnostic applications of hydrogel-forming MNs have also been described for the detection of glucose [58] and lithium monitoring [59]. Fine-tuning of the delivery time is possible by adjusting the polymer decomposition from minutes to days. Nonetheless, their low strength and limited drug doses are among the limitations of hydrogel-forming MNs. This approach requires dramatic improvements to be used for any feasible commercial applications in the near future.

## 3. Challenges of the Microneedle Delivery System

Endorsing the translation of MNs from research laboratories to the relevant industries is an exciting but demanding task for the near future. To translate this innovative technology from the lab bench to feasible products in the relevant markets, some crucial questions and challenges should be considered promptly. We hereafter discuss these challenges and active strategies to address these difficulties, which could determine the future of the field and its commercial applications. The main issues/concerns for the development of a microneedle-based delivery system is summarised in Figure 3 and discussed in the following sections.

### 3.1. Parameters Affecting MN Insertion

The capability of MN patches to adequately puncture the skin is a vital requirement. When addressing this matter, the skin’s characteristics, which might vary across the body and vary from person to person, should also be taken into account. The insertion and penetration behaviour of MNs to overcome the skin’s elasticity is strongly dependent on several parameters, such as geometry, base and tip diameters, length, and interspace (centre-to-centre spacing) [60,61]. An approach of “one-size-fits-all” cannot be envisaged in any design and development stages for any MN application. Infiltration and active delivery performance of MNs are strongly related to the geometry of individual MNs and the array, MN materials, the MN management method, and the characteristics of skin tissue [62]. Depending on the target medicines and applications, the microneedle mechanical strength, insertion depth, and drug release profile could be finely tuned by modifying the microneedle shape and composition.

*The geometry*: The geometry of MNs is a parameter that should be taken into consideration early on when developing MNs for clinical applications. A recent study indicated that the mechanical strength and penetration characteristics of MNs are affected by the geometric structure of microneedle arrays [63]. Simulations have shown a linear relationship between the mechanical strength and the number of vertices in the polygon base (e.g., triangular, square, and hexagonal microneedle bases), showing better insertion depths for the triangular and square-built microneedles. Superior capacity to insert into the skin was observed for the sharper edges of the triangular and square MNs compared to the hexagonal MNs. In a recent study, cone-shaped MNs were discovered to possess the ideal geometry for the delivery of ovalbumin and transcutaneous immunisation, with both greater needle insertion and a fast dismantling time for a more potent immune response obtained [64]. A further improvement has been proposed recently to reduce the risk of insufficient drug delivery, wherein an array of hemispherical convexities was positioned in the lower half of the cone-shaped dissolving MNs to increase drug flux [65].

*Tip diameter and Sharpness*: Tip diameter is another parameter for MN insertion. Relatively blunt MNs (tip diameters of 60–160 µm) require a relatively high insertion force (0.08–3.04 N) for controlled applications of MNs and are linearly reliant on the tip frontal area [66]. To achieve a well-controlled manner to the desired depth, the fabrication of MNs with sharp tips is essential. For the successful delivery of therapeutics, it has been reported that MNs with smaller tip diameters (<15 µm) access the skin more smoothly than MNs with a tip diameter of larger magnitude. This is particularly important in vaccine delivery to achieve appropriate control over the penetration depth of MNs, not only for delivering the antigens but also for specifically targeting Langerhans cells residing in the epidermal layer or dendritic cells dwelling in the dermal layer of the skin for a robust adaptive response [67]. The sharpness of the tips of microneedles can aid and control the puncture force. An increased tip sharpness, however, not only reduces the puncture force but also reduces the structural strength of the microneedles, leading to a high risk of breakage.

*Application velocity and force*: In close relationship with the tip diameters, the application velocity and force are other parameters in the MN delivery system that should be considered in detail. Several studies have reported that the penetration depth of MN arrays varies (from 10% up to 80%) and increases with the application velocity and force [68]. A variety of patch configurations have been used, with similar outcomes of the penetration force per microneedle obtained. A 25-microneedle array with a tip radius of <100 nm requires an insertion force of 10 mN per microneedle for effective penetration into the skin [69]. Two independent studies have also acknowledged these findings and reported that insertion forces of 15–20 mN [70] and 15–30 mN [71] per microneedle were required for operational insertion. These forces represent arrays of 10–100 microneedles, which give a total applicator force of 0.1–3 N. Although these forces are low, the need for consistent application may necessitate a controlled application approach or a device.

*Length:* Because the thickness of the SC and other skin layers differs across individuals, the particle insertion depth may also vary. The transport capability of the skin, once a MN patch has been applied, will depend on the perforation depth of the tissue. If a drug is relatively small and has high diffusion capacity, creating surface pores by microneedle application should be sufficient for therapeutic function. However, if rapid delivery to the bloodstream is the goal, it may be preferable to create pores that reach the dermis, where capillaries are located. This may be one reason for assorted microneedle lengths that have been reported to date. In addition to the shorter microneedles, there have been many studies that used long microneedles (up to 1000 μm long) to increase insulin permeability into the skin [72].

*Interspace (centre-to-centre spacing):* The skin is a topographically diverse surface capable of withstanding significant deformations prior to penetration. A significant number of distinct punctures must be generated when there is a high-density array of microneedles (e.g., more than 500/cm^2^). This takes a lot of energy. Naturally, as the density and number of microneedles grow, so does the necessary force for skin puncture. This can result in increased feeling for the patient and may require the use of a larger/stronger device for certain applications. Needles with increasing width, length, and density can result in larger, longer, and more crowded holes, through which a higher amount of medication may diffuse. However, more tightly placed needles may cause the “bed-of-nails” effect too [68].

### 3.2. Biocompatibility, Biodegradability, and Stability

One of the safety aspects of MN systems in clinical use is biocompatibility. To ensure that MN products are acceptable for human exposure, several tests are required to evaluate their biocompatibility based on contact periods of less than 24 h, between 24 h and 30 h, and more than 30 h. [73]. For the former two periods, the corresponding tests are cytotoxicity, sensitisation, irritation, and intracutaneous reactivity tests. Genotoxicity and subacute/subchronic systematic toxicity tests are additionally recommended for the latter period of use. The use of biodegradable materials is desirable for microneedles because these materials can be degraded and removed from the body safely. Therefore, using biodegradable polymeric systems for MN fabrication has been pursued in recent years. The primary benefit of polymeric microneedle systems is their ability to load medication into the microneedle matrix for discharge in the skin via biodegradation or dissolution in the body fluid of the skin.

The ability to manufacture microneedle structures from aqueous polymeric mixtures at room temperature without the requirement of a heating step might be a significant benefit in retaining the stability of an integrated medication, particularly in the case of therapies in which proteins and peptides are involved. Nonetheless, the stability of MN cargo has to be evaluated to ensure that fragile and easily degradable therapeutics are protected during storage. This is usually done by storing MNs and their cargo at various temperatures, including −25 °C, 4 °C, 20 °C, 40 °C and 60 °C, followed by analytical assessments. Generally, the protein cargo of MNs has better storage stability and longer shelf-life due to the rigid glassy microneedle matrices restraining the molecular mobility and limiting access to atmospheric oxygen [74]. This can be further extended by the incorporation of stabilisers, including trehalose and sucrose. Attention to water is particularly critical when non-vacuum storage conditions are present, as they can not only destroy the stability of laden cargo but also the mechanical properties of the MNs themselves [75]. Dissolvable MNs are very susceptible to the surrounding humidity; therefore, the storage environment should be dry and cool for prolonged stability and extended shelf-life.

### 3.3. Loading Capacity and Dosage Accuracy

*Loading capacity:* A coated microneedle device can only deliver a bolus dose of around 1 mg of medicine. Although hollow microneedles allow for continuous infusion or “as-needed/on-demand” dosing, central exits may be obstructed by compressed skin tissue after microneedle insertion. Even though MNs have the potential to overcome the skin’s barrier properties, their success is very much dependent on passive diffusion of the biological formulation into the skin. This can make it difficult to administer large dosages, and much of the dose can be lost on the skin’s surface. As a result, the time of application and the inability to monitor dose delivery have caused reluctance to use this technology for certain clinical applications. One example is the distribution of vaccines for which dosage constancy is critical. Recent work has shown that administering vaccines directly to the epidermis and dermis of the skin has the potential to induce immunological responses with considerably less vaccine than standard intramuscular injection. These advantages, however, might be lost if just a tiny fraction of the administered dosage reaches the skin. While this is not an insurmountable obstacle to this technology, vaccines, in particular, require a threshold dosage to induce immunity, which might be more difficult to achieve when depending on passive diffusion.

*Dosage accuracy*: The dosage accuracy of MN delivery systems in continuous drug delivery is an issue that requires close attention. Several methods using separable microneedles have been proposed for minimising the patch-wearing time and quickly removing the formulation from the MNs [76,77]. Storing and delivering protein drugs, including insulin, erythropoietin, glucagon, growth hormones, and parathyroid hormones, are challenging tasks, as bio-macromolecules are prone to quick degradation and inactivation. These matters could be best handled by not only the incorporation of stabilisers but also by considering the whole process of MN manufacturing parameters, such as manufacturing and storage temperatures and drying conditions, polymer concentration, sterilisation, and packaging. As discussed earlier, MNs can be manufactured in various types and materials. The drug delivery efficiency when using solid MNs is rather difficult to control accurately. Coated MNs can efficiently deliver precise amounts of a drug but have limited drug loading capacity due to their small surface area for coating. Encapsulating drugs in the matrices of MNs is possible if dissolvable microneedles are fabricated primarily from hydrophilic, biocompatible, and biodegradable materials, and if the cargo can be discharged entirely within the skin’s interstitial fluid without leading to unwanted debris. Relatively large doses and the controlled release (slow or fast delivery) of various drugs can be transferred without issues of reservoir leakage. Dissolvable microneedles might be an efficient approach to preserve and stabilise nano-sized compositions while improving nanoparticle penetration through the stratum corneum barrier. Various approaches have been thoroughly studied, and several analytical techniques for tracing and tracking the journey of nanomaterials with their valuable payloads, both in vitro and in vivo, have been developed [78,79].

### 3.4. Skin Irritation and Recovery

The immunogenic nature of the skin makes it a highly responsive organ towards the MN delivery of any therapeutic agent. Mild and temporary erythema may develop as a side effect depending on the size, substance, and type of the given medication. Skin irritation, sensitisation, and immune response must also be evaluated as part of the safety assessments of MN products during clinical trials. This safety concern must be evaluated using animal testing before any human clinical trials. On the other hand, great immune responsiveness of the skin may present an opportunity for MN-based vaccine delivery if other obstacles have been addressed properly, as discussed.

### 3.5. Cost of Microneedle Fabrication

Current microneedle manufacturing processes need to be improved to reach large-scale production in order to completely transfer microchip-based microneedles into therapeutic applications. Until now, extensive economic evaluations of the technology have not yet been quantified thoroughly, but it is not difficult to predict that, as with every new technology, the clinical use of MNs can be comparatively expensive due to the complex fabrication and storage procedures and the slow and long approval process.

Even though MNs show promise in preclinical research, their economic and epidemiologic implications have yet to be assessed thoroughly. However, to give an idea of the costs involved in MN application in vaccine delivery, one study projected the hypothetical costs of MN-based measles vaccine injection and compared them with those of traditional subcutaneous (SC) injection [80]. For a population of 1 million children, the estimated prices of a 2-dose vaccination program using the microneedle patch and SC injection were USD 0.95 (range USD 0.71–USD 1.18) for the first dose and USD 1.65 (range USD 1.24–USD 2.06), respectively, assuming that the MN vaccine method is more heat-stable and requires cost-effective cool chains. The total costs of the vaccination program were estimated to be USD 1.5 million for MN-based administration compared with USD 2.5 million for SC administration. The authors commented that the cost-effectiveness of MN patches depends on numerous factors, including approval rates and the effectiveness of the MN patches in relation to the traditional subcutaneous vaccine delivery method [80]. In another study, an economic model was applied to assess the value of MN patch technology for the seasonal influenza vaccine [81]. The model predicted that its introduction would be economical or dominant at a USD 9.50 price point in the majority of situations evaluated when healthcare workers managed the MN-based system. If efficacy rose by ≥3%, MN vaccination would be cost-effective or dominant for all price points ≤ USD 30 for all administration scenarios studied. The growing economic pressure on the global healthcare system makes it crucial for researchers to study the costs related to MN-based delivery systems. While the technology has great potential for transdermal drug delivery, its success is very much dependent on carrying out economic assessments while the technology is still under development, as should be the case for the development of any new science-led, research-driven product developments. Along with the other factors discussed earlier, the success of any MN application is also very much dependent on the fabrication technique and materials used. Scaling up to industrial manufacturing for mass production necessitates a focused strategic plan. Several factors should be considered carefully, including the accurate and reproducible production of MNs (attributed to the maturation of MN fabrication technology), the ability to expand for mass production, and the expandability of MN technology to a wide range of concerns or diseases, together with regulatory approval and clinical adaptation [82]. The choice of materials is also of utmost importance and should be compatible with the laden cargo for optimal insertion and delivery performance without any deleterious effects on the bioactivity and stability/viability of the therapeutic agents. For optimisation, an ideal production method should aid easy, rapid, and cost-effective modifications in the material and geometry parameters.

### 3.6. Sterilisation of the Microneedle Patches

MN patch sterilisation is another challenge that should be taken into account early on when MN-based products are aimed for commercial application. If sterilisation is necessary, then the method of choice will be critical, because the most widely used methods, such as moist heat, gamma or microwave radiation, and ethylene oxide may deleteriously affect any cargoes with sensitive ingredients, including biomolecules, vaccines, peptides, and/or even the microneedles themselves [83]. Although the risk of introducing bioburden into the sterile area of the body (e.g., epidermis and dermis) by MNs is significantly smaller than a single puncture by a hypodermic needle, complete sterilisation of MNs-based products may be obligatory by the regulatory bodies to safeguard the users. The material used for MN fabrication determines the method of choice for sterilisation. For solid MNs of metals, silicon, and glass, the sterilisation is straightforward; dry heat sterilisation, moist heat sterilisation, and gamma radiation are the most common methods employed [84]. However, when delivering fragile biological active ingredients is in demand (e.g., using coated MNs), the method of choice should be carefully evaluated in terms of maintaining stability and activity of the coated ingredients. MNs constructed by carbohydrates and polymers (e.g., dissolving MNs) present the biggest challenge when choosing the sterilisation method, since the sterilisation not only affects the fragile loads but also the morphological, physicochemical, and mechanical properties of MNs themselves. The effects of various sterilisation methods, such as moist and dry heat sterilisation and gamma radiation, on dissolving and hydrogel-forming MNs have been studied, with ibuprofen and ovalbumin as model drugs [85]. It was found that no measurable bioburden was detected, and levels of endotoxin were under the FDA limits if aseptic preparation was followed. However, moist and dry heat sterilisation damaged all formulations, whereas the gamma irradiation at a sterility assurance level (SAL) of 10^−6^ (according to the British Pharmacopeia) can be used for sterilisation without causing structural damages or affecting delivery capabilities of hydrogel-forming MNs. The radiation, however, destroyed ovalbumin and changed the appearance of ibuprofen. Alternative methods for delicate MNs have been proposed [86,87]. Ethylene oxide and electron beam sterilisation were shown to be effective but less destructive methods for MN sterilisation. In another study, a self-sterilisation of MNs was proposed, in which silver nanoparticles were embedded in CMC MNs. The authors implied that the pores produced by MNs were free from microorganisms until the skin is healed completely [88].

It is clear that the information available in the literature is rather limited, and therefore, the sterilisation of MN-based products requires extensive research before going into commercial production and approval; this presents one of the most important challenges in MN-based delivery systems. In particular, endpoint sterilization for MN products requires a great deal of attention, as MN manufacturing under an aseptic condition could be both complicated and costly.

### 3.7. Regulation of the Microneedle Patches

The quality of submissions received from combination products employing microneedles has been a source of concern for the US FDA, particularly in the areas of stability testing, content consistency, risk analysis, sterility validation, and manufacturing. As discussed earlier, MNs are a viable option for the delivery of therapeutic agents such as hormones, vaccines, enzymes, mRNA, and difficult-to-deliver small molecules via the skin. In view of the regulatory body, clinical application scenarios, as well as the repeatability and efficacy of microneedle devices, should be thoroughly shown using cell studies, animal testing, and clinical trials. Furthermore, a thorough understanding of human physiological settings, thorough examination of clinical demands, and the mobility and simplicity of microneedle devices can all help to promote such clinical translations. The number of MN-based medicinal products for therapeutic applications is rising exponentially. However, the submission process to the FDA for approval is not straightforward because submissions should be in the form of combination products that use microneedles. Any submissions of this kind require satisfactory information about product analysis, testing and validations such as risk analysis, content uniformity, stability testing (formulation/API migration/mechanical characteristics), sterility validation, and manufacturing. The FDA has stated “Regulation of combination products must take into account the safety and effectiveness questions associated with each constituent and the product as a whole” [73]. The current strategy of product-specific approval (rather than specific MN-systems) for the licensing of microneedle products adopted by the regulatory bodies causes great delays in approval, thereby restricting the commercialisation of MNs. To promote the commercialisation of MN products, the cGMP and quality control should be merged, and licencing regulations must be defined clearly, covering the shape, formulation, sterilisation, and packaging. On the other hand, the clinical development of microneedle devices can advance separately to drug or vaccine formulation. This can greatly simplify any regulatory processes and might allow for more rapid incorporation of the technology into the supply chains of particular drugs. Although just small quantities of a molecule may reach their intended delivery sites, this approach’s simplicity may lessen the regulatory problems encountered by other complicated formulation techniques.

Due to the possibility that this MN design may be CE marked as a medical device rather than a medicinal product, pharmaceutical firms may be prepared to invest in such a device before investing in medication that incorporates MNs. In addition, robust guidance is required to fully classify MN-based products; nevertheless, it has been proposed that this would most likely come within the medical device category for monitoring/diagnostic applications, and as a “combination product” (drug and device) or “drug product” for the delivery of drugs or vaccines [89,90]. Once this distinction is made, it may be possible to adapt existing quality control procedures for MNs. The current standard quality control methods may not be completely suitable to MN products due to the inherent differences between transdermal patches and hypodermic needles. If all remaining concerns can be suitably addressed to meet the needs of both regulators and patients, the goal of bringing MN-based products to the transdermal market will soon become a reality. In 2020, the first new drug application for a pharmaceutical microneedle patch, Qtrypta, was submitted to the Food and Drug Administration (FDA) by Zosano Pharma. The patch is a titanium microneedle with a coated zolmitriptan for acute migraine treatment.

## 4. Conclusions and Perspectives

Development of marketable microneedle-based drug delivery products is highly likely in near future. Extensive research in MNs is being conducted for the efficient delivery of therapeutics, as innovative transdermal drug delivery methods are urgently required to expand the transdermal market for hydrophilic molecules, macromolecules, proteins, and conventional medicines for new therapeutic indications. The future of the microneedle industry seems to be quite bright, with the rapid realisation of new information fuelling industrial progress. The effectiveness of MNs has been demonstrated in several clinical trials, but there have still been far more preclinical studies. Experts from academia, industry, and regulatory organisations are collaborating to help MNs to advance into safe and effective clinical usage provided that the shortcomings associated with these systems are promptly and rationally addressed. It is believed that, in time, microneedle-based technology will lead to improved illness prevention, diagnosis, and control, as well as an increase in the health-related quality of life of patients globally. Nonetheless, the complicated and expensive production of MNs, together with several application-related difficulties, could delay their clinical translation. This is evident from the clinical translation of microneedle applications in the pharmaceutical industry. For instance, the lack of clinical data on “www.clinicaltrials.gov” using “microneedle vaccine” indicates that the scale-up production of MNs is still a challenge. What is more, novel manufacturing methods, micromachining and 3D printing technologies in particular, are envisaged to lower the costs and simplify fabrication procedures in the near future.

## Figures and Tables

**Figure 2 micromachines-12-01321-f002:**
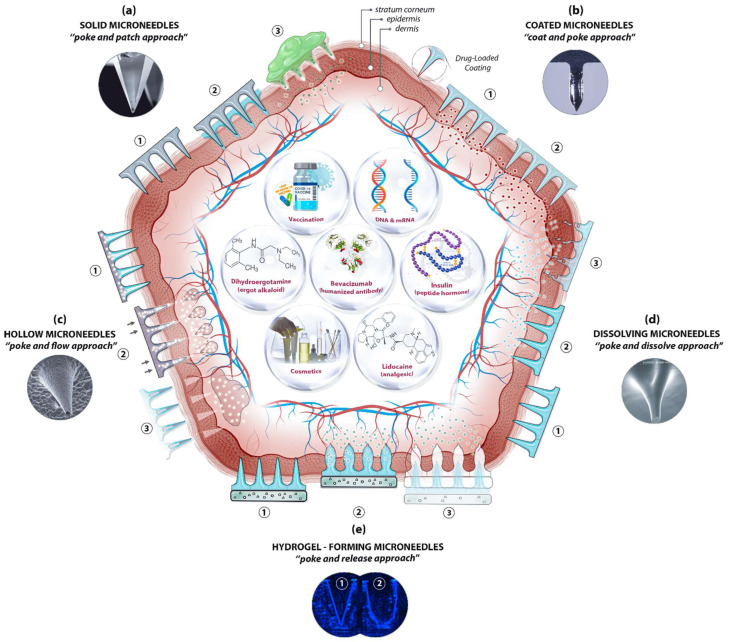
A schematic diagram of microneedle (MN)-based drug delivery approaches with the cross section of the upper layer of the skin. The approaches are (**a**) solid MNs, (**b**) coated MNs, (**c**) hollow MNs, (**d**) dissolving MNs, and (**e**) hydrogel-forming MNs. The step-by-step process of each delivery approach is numbered from 1 to 3. Representative microscopic images of MN types and examples of deliverable payloads such as drugs and bio-macromolecules are also shown. Images was adopted with permission from [42,43]).

**Figure 3 micromachines-12-01321-f003:**
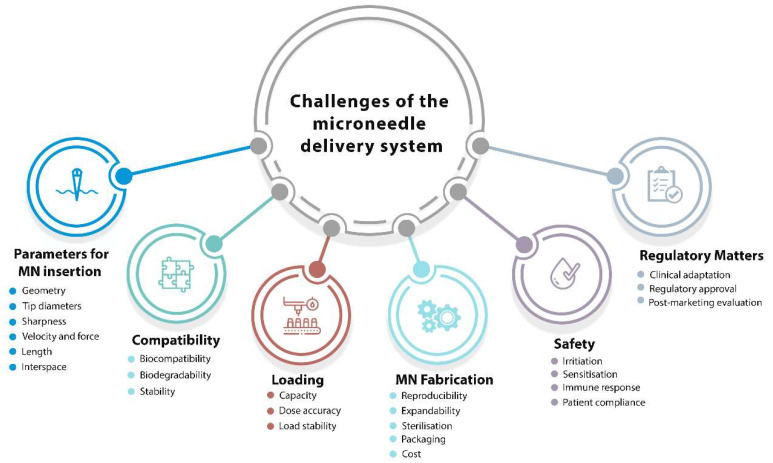
Factors effecting development of microneedle-based delivery system.

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
