# Peer review of "Microneedles in Drug Delivery: Progress and Challenges"

_micromachines, 2021, doi:10.3390/mi12111321_

Round 1

Reviewer 1 Report

The review focuses on challenges associated with microneedle scale-up and clinical translation, which sets it apart from the many microneedle based reviews published recently. However, here are some points of note:

  1. The English language needs to be refined. Sentence structuring is poor with many sentences being grammatically incorrect.
  2.  Figure legends can be descriptive, esp for Figure 2, which is hard to understand without figure description in the legend or in the main body.
  3. Sterility is a major problem with microneedle scale-up. While the authors mention this problem, a better discussion including sterilization procedures and relevant studies should be included.
  4. The manuscript lacks any tabular content. It would be easier to read information provided in, for example, "Parameters affecting MN insertion" if it is also presented in a tabular format.

Reviewer 2 Report

The present manuscript investigates the progress and challenges of using microneedles for drug delivery. The manuscript is well written and well organized. On the other hand, there are several recent review articles in this topic. It should be noted that the challenges of Microneedle delivery system is covered in a separate section. Keeping in mind that this is a review article that mainly discusses the possible limitations of the Microneedle method, the literature review is poor. Therefore, I suggest the authors to add additional references to research articles in the challenges section.

Some representative but not colte list is the foloowing

1. Transdermal Delivery of Drugs with Microneedles—Potential and Challenges, Pharmaceutics 2015, 7, pp 90-105.

 2. Recent Advances and Challenges in Microneedle-Mediated Transdermal Protein and Peptide Drug Delivery, Biomaterials and Bionanotechnology Advances in Pharmaceutical Product Development and Research 2019, pp 495-525.

 3. An optimized method for 3D magnetic navigation of nanoparticles inside human arteries, Fluids 6(3), 2021, 97.

4.  Dissolving microneedles for transdermal drug delivery: Advances and challenges, Biomedicine & Pharmacotherapy, 93, 2017, pp 1116-1127.

  5. On the magnetic aggregation of Fe3O4 nanoparticles, Computer Methods and Programs in Biomedicine, 198, 2021, 105778.

Consequently, the manuscript should be reconsidered for publication in Micromachines journal after major revision.